# FOCT: Few-shot Industrial Anomaly Detection with Foreground-aware Online Conditional Transport

## ABSTRACT

Few-Shot Industrial Anomaly Detection (FS-IAD) has drawn great attention most recently since data efficiency and the ability to design algorithms for fast migration across products have become the main concerns. The difficulty of memory-based IAD in low-data regime primarily lies in inefficient measurement between the memory bank and query images. We address such a pivotal issue from a new perspective of optimal matching between features of image regions. Taking the unbalanced nature of query features into consideration, we adopt Conditional Transport (CT) as a metric to compute the structural distance between representations of the two sets to determine feature relevance. CT distance generates the optimal matching flows between unbalanced structural elements that achieve the minimum matching cost, which can be directly used for IAD since it well reflects the differences of query images compared with the normal memory. Realizing the fact that query images usually come one-by-one or batch-by-batch, we further propose an Online Conditional Transport (OCT) by making full use of the current and historical query images for IAD via simultaneously calibrating the memory bank using the online query images and matching features between the calibrated memory and the current query image. Go one step further, for sparse foreground products, we employ a predominant segment model to implement Foreground-aware OCT (FOCT) for improving the effectiveness and efficiency of OCT by forcing the model to pay more attention to diverse targets rather than redundant backgrounds when calibrating the memory bank. FOCT can improve the diversity of calibrated memory during the IAD process, which is critical for robust FS-IAD in practice. Besides, FOCT is flexible since it can be friendly plugged and played with any pre-trained backbones, such as WRN, and any pre-trained segment models, such as SAM. The effectiveness of our model is demonstrated across diverse datasets, including benchmarks of MVTec and MPDD, achieving SOTA performance.

## CCS CONCEPTS

• **Computing methodologies** → **Scene anomaly detection**.

## KEYWORDS

Industrial Anomaly Detection, Few-shot Learning, Foreground-aware Online Conditional Transport

**Unpublished working draft. Not for distribution.**

## 1 INTRODUCTION

The fragmented nature of industrial anomalies, such as subtle bruises and obvious breakages with various appearances and scales [24, 28, 40], raises difficulties in detecting and classifying industrial anomalies in the fully-supervised manner as primal researches do [9, 11]. Therefore, unsupervised Industrial Anomaly Detection (IAD) methods have been developed most recently, which efficiently use the model trained with only normal industrial images of each product to detect anomalous industrial images and localize the corresponding anomalous regions precisely [3, 4, 17, 23].

Researches on unsupervised IAD can be divided into two categories, namely reconstruction-based models [25, 26, 42] and memory-based models [10, 27, 40]. Reconstruction-based models aim to learn continuous or discrete feature representations of normal images utilizing Deep Generative Models (DGMs) such as Variational AutoEncoding (VAE) [20, 42] and diffusion models [13, 25] through a cumbersome training process. Memory-based models usually extend DGMs with a memory bank to record the normal features given normal training images, manifesting in various forms, including CNN-based multi-scale features [10, 28] and Transformer-based learnable query embeddings [40]. At test time, feature matching between test images and memory is preferred. However, both lines of work heavily rely on massive normal training images, either for DGMs optimization or memory learning, and fail to generalize across products quickly in low-data regime.

Inspired by how human beings detect anomalies, Few-Shot Learning (FSL) [35, 36] is introduced to IAD for learning a common model shared among multiple products and also generalizable to novel products where limited normal training images are provided, for example, 1 or 2 shot per product. This new paradigm is also known as Few-Shot IAD (FS-IAD) [8, 14, 15, 31], which can be divided into Inductive FS-IAD (IFS-IAD) models [14, 15, 31] and Transductive FS-IAD (TFS-IAD) [8] models. The former leverages statistics of limited support (training) images to conduct IAD, while the latter uses statistics of both support and query (test) images for more generalizable IAD. TFS-IAD and IFS-IAD models are separately developed to extend the generalization capability, mainly for reconstruction-based and memory-based models of the previous studies. Additionally, meta-learning based IFS-IAD models [14] have also been proposed to realize fast generalization to new products.

Despite their promising results, we observe that redundant backgrounds and large appearance variations of support images may drive the image-level embeddings from the normal pattern far apart in a given metric space at test time. Although the issue could be alleviated by Deep Neural Networks (DNNs) under data-hungry supervised training, it is almost inevitably amplified in FS-IAD and thus negatively impacts anomaly detection and localization. Moreover, employing heuristic and linear metrics for anomaly detection [8, 31] destroys image structures and loses local features, which can

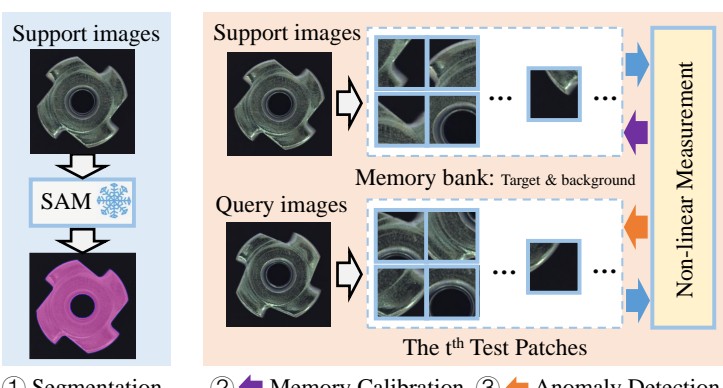 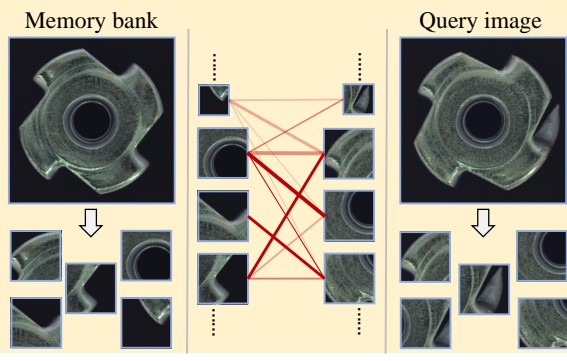

① Segmentation ② ◀ Memory Calibration ③ ◀ Anomaly Detection ④ Optimal Matching Flows

**Figure 1: Illustration of our proposed framework for FS-IAD task. We first use pre-trained prompt-guided SAM for support image segmentation. Then we design a novel online learning pipeline for memory calibration and anomaly detection, simultaneously, which are implemented using the optimal matching flows between the memory and query images.**

provide discriminative and transferable information across normal and anomalous patterns. It should be critical for IAD in the few-shot scenario. Therefore, a desirable metric-based algorithm should have the ability to leverage the local discriminative representations for feature matching and minimize the impact caused by the irrelevant regions for robustly distinguishing anomaly images and their corresponding regions. Intuitively, we hope to observe that an anomalous region should be most irrelevant to the space spanned by normal features. Furthermore, how to discover the characteristics of IAD itself and absorb them into a metric-based algorithm design for improving generalization is another crucial problem to be contemplated.

We aim to solve the pivot issues mentioned above under the line of memory-based TFS-IAD, which was rarely studied before. First of all, considering the fact that query features exhibit unbalanced distribution along the memory bank, as detailed in the methodology later, we adopt a theoretically guaranteed non-linear measurement function called Conditional Transport (CT) to compare the building blocks of two complex structured representations. CT [43] is a function for computing distance between structural representations, which was originally proposed for GAN-based DGM optimization, and can be friendly cascaded with DNNs for feature matching, such as WRN [41] in this work. Given the distance between all element pairs, CT has the formulation of the transport problem [12, 37] and the optimal matching flows between two structures can be achieved by minimizing the matching cost via Stochastic Gradient Descent (SGD). Secondly, previous researches ignore the fact that industrial images to be detected are usually coming online, which provides a hotbed for enhancing the capability of CT function with both current and historical query images. Therefore, we initialize a memory bank with support features and propose an online learning pipeline to calibrate the memory with statistics of online query images, the anomaly detection is thereby implemented by measuring the distance between the calibrated memory and query features with the CT function. We name this enhanced model Online CT (OCT). Thirdly, we find that distinguishing products and their backgrounds in the query images is necessary to realize robust FS-IAD for sparse foreground products. On the one hand, redundant backgrounds

are not conducive to OCT, which may introduce overwhelming invalid calculations. On the other hand, a well-calibrated memory with a proper foreground and background ratio performs steadily in FS-IAD. We use a predominant segment model, such as SAM [21] in this work to implement Foreground-aware OCT (FOCT) to improve the effectiveness and efficiency of OCT by forcing the model to pay more attention to diverse targets rather than redundant backgrounds. We sum up our proposed framework in Fig. 1. Our work contributes in the following ways:

- We propose a novel memory-based TFS-IAD framework that primarily adopts a non-linear and theoretical guaranteed CT function between the memory bank and query images for precise feature matching.
- We make full use of the characteristics of IAD and design a new pipeline called FOCT to enhance the generalization capability of the aforementioned CT function.
- Our proposed model achieves SOTA anomaly detection and localization performances in the few-shot regime under various datasets, including MVTec [1] and MPDD [16] and fruitful settings, including 1/2/4-shot.

## 2 RELATED WORK

### 2.1 Industrial Anomaly Detection

IAD involves handling industrial training datasets that exclusively consist of normal data, presenting challenges due to the varying subtleties of defects. In this field, the prevailing unsupervised techniques are predominantly reconstruction-based methods [10, 38, 40] and memory-based models [5, 7, 28]. Reconstruction-based methods are trained exclusively with normal data on the premise that anomalies will yield significantly higher reconstruction errors. Nevertheless, this assumption does not invariably apply, occasionally leading these approaches to encounter the identical shortcut issue, where they inadvertently learn to reconstruct anomalies, thus reducing their discriminative efficacy. Recently, memory-based models have achieved promising performances in anomaly detection and localization, leveraging pre-trained features stored in memory banks containing various feature hierarchies, such as PaDiM [7],

SPADE [5], Patchcore [28], thereby significantly enhancing the detection capabilities. Furthermore, various sophisticated adaptations have been implemented, including normalizing flows [19, 29] and student-teacher knowledge distillation [2, 30] integrated with the features obtained from pre-trained networks, to effectively manage the discrepancy in distribution between the pre-training natural image datasets and the distinct nuances present in industrial imagery. This paper distinguishes from the previous studies by concentrating on few-shot anomaly detection, where only a limited number of normal images are accessible.

## 2.2 Few-shot Industrial Anomaly Detection

FS-IAD has emerged as a compelling new area of research, aiming to identify anomalies with only a few support samples from target categories. FS-IAD methodologies can generally be categorized into two types: Inductive FS-IAD (IFS-IAD) models [14, 15, 31] and Transductive FS-IAD (TFS-IAD) models [8]. The majority of existing research has focused on inductive FS-IAD approaches, which utilize statistical information from a small set of support (training) images, without incorporating query (test) images to perform anomaly detection. Within this realm, generative models have gained popularity recently, including the hierarchical generative model TDG [33], and the normalizing flow-based model DiffNet [29], which both sketch out data distributions from the limited support samples. Furthermore, the recently proposed Opt-PatchCore [31] introduces an effective data augmentation method that substantially enhances feature diversity, thus boosting the generative capabilities of the model's memory bank. Concurrently, a feature-augmentation strategy embodied by GraphCore [39] incorporates visual isometric invariant features into the memory-based anomaly detection framework, thereby improving its ability to distinguish anomalies. In contrast, TFS-IAD models enhance anomaly detection's generalizability by utilizing statistical data from both support and query images. An innovative approach is demonstrated by Fastrecon [8], which employs a regression technique with distribution regularization. This technique achieves the optimal transformation from support to query features, ensuring the reconstruction closely resembles the query sample while preserving the characteristics of normal samples. However, the reliance on heuristic and linear metrics for anomaly detection can compromise image structure integrity and overlook the importance of local features. In this work, we introduce a novel memory-based TFS-IAD framework that leverages Conditional Transport (CT) as a metric to calculate the structural distance between the memory bank and the query images, thereby enhancing the generalizability of FS-IAD.

## 3 PRELIMINARY

In this section, we introduce the preliminaries on the task formulation of FS-IAD and the theory of CT function to prepare for subsequently presenting our proposed model, FOCT.

## 3.1 Task Formulation

We formally define the one-class industrial anomaly detection task following the standard $n$ way $k$ shot few-shot learning setting. In each category, there are $k$ normal support images $x^s_{1:k}$ to re-train or fine-tune the model. At test time, given the current query image

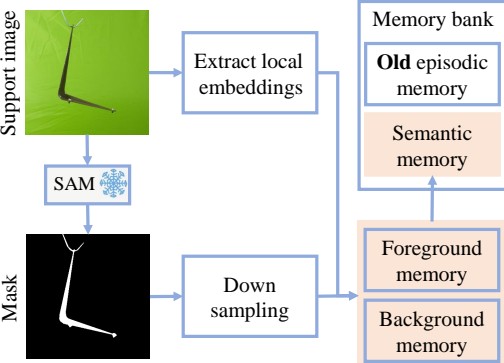

**Figure 2: Illustration of using SAM to build the semantic memory that is aware of foreground and background contexts for sparse foreground products.**

$x^q_t$ being whether normal or anomalous and its historical query image series $x^q_{1:t-1}$ of the corresponding category as $x^s_{1:k}$ do. The model predicts whether or not the query image $x^q_t$ is anomalous at the pixel and image level based on the query image series $x^q_{1:t}$. It is worth mentioning that previous methods only leverage statistics of the current query image $x^q_t$ to make predictions while they always ignore the statistics of the historical query image series $x^q_{1:t-1}$, which may yield some suboptimal results as revealed by our work.

## 3.2 CT Theory

Conditional Transport (CT) is proposed to measure the distance between two sets of weighted objects or distributions using the non-linear mapping function such as DNN, which is built upon the basic distance between individual objects. Specifically, given two sets of discrete sampled objects $y_{1:n}$ and $z_{1:m}$ from distributions $p(y)$ and $p(z)$, where $p(y) = \sum_{i=1}^{m} a_i \delta_{y_i}$ and $p(z) = \sum_{j=1}^{n} b_j \delta_{z_j}$. $a \in \Delta^m$ and $b \in \Delta^n$ separately denote the probability simplex of $\mathbb{R}^m$ and $\mathbb{R}^n$. The CT distance is defined as:

$$\mathcal{L}_{\phi,\rho} = \min_{\phi} \sum_{y \in p(y)} \sum_{z \in p(z)} \rho C_{\phi}(y \to z) + (1-\rho)C_{\phi}(z \to y) \quad (1)$$

where $C_{\phi}(y \to z)$ and $C_{\phi}(z \to y)$ separately denote the forward and backward CT distances, also known as navigators in the scope of CT theory, $\phi$ is optimizable parameters, and $\rho$ is a trade-off factor controlling strengths between the forward and backward navigators. To better understand how CT works when minimizing Eq. 1, we take the forward navigator as an example as follows:

$$C_{\phi}(y \to z) = \sum_{i=1}^{m} \sum_{j=1}^{n} \frac{1}{n} \pi_n(z_j|y_i;\phi)c(y_i,z_j)$$

$$\pi_n(z_j|y_i;\phi) = \frac{exp(-d_{\phi}(y_i,z_j))}{\sum_{r=1}^{n} exp(-d_{\phi}(y_i,z_r))} \quad (2)$$

where $c(\cdot,\cdot)$ is Euclidean distance, $\pi_n(z_j|y_i;\phi)$ represents forward transport matrix satisfying $\sum_{j=1}^{n} \pi_n(z_j|y_i;\phi) = 1$, it depicts the probability of transporting $y_i$ to $z_j$. $d_{\phi}(\cdot,\cdot)$ denotes non-linear measurement function, such as MLP implemented in [43]. The forward CT can be interpreted as the expected cost of stochastically

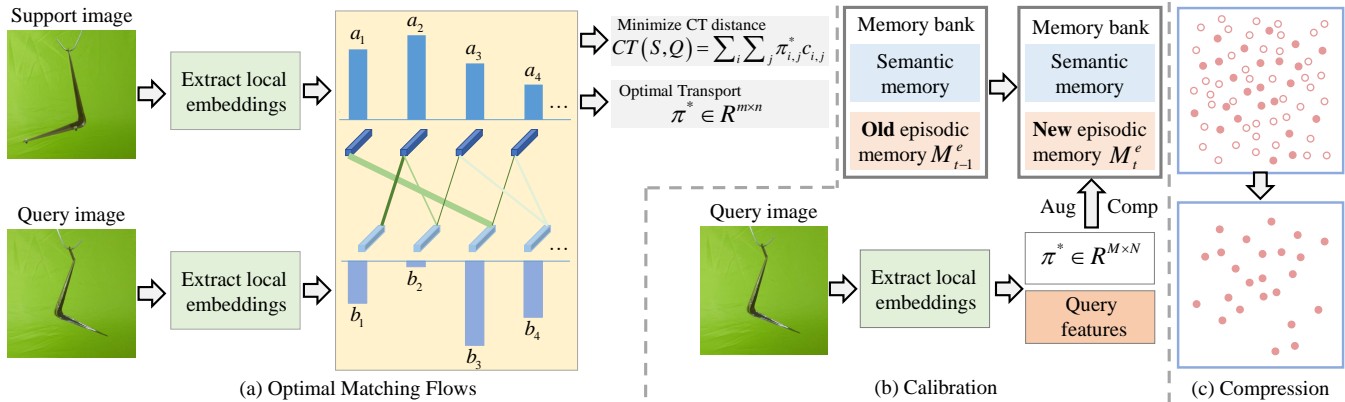

(a) Optimal Matching Flows          (b) Calibration          (c) Compression

**Figure 3: (a) Optimal matching flows calculated between the support and query features. (b) The pipeline of episodic memory calibration consists of memory augmentation and memory compression. We use the optimal transport matrix $\pi^*$ derived from the optimal matching flows to select the Top-K query features for augmenting the episodic memory. We employ the coreset technique to downsample the episodic memory for compression, as shown in (c).**

transporting a random source point to one of the $m$ randomly instantiated "anchors" of the target distribution. By minimizing the forward CT of Eq. 2, an unbiased statistical estimation of the two distributions arrives. Hence, it is reasonable to measure the relevance of one distribution against another.

## 4 METHODOLOGY

In this section, we outline the details of our proposed FOCT model from the following perspectives: We first use pre-trained WRN [41] and SAM [21] to extract foreground-aware support features preparing for constructing the semantic memory (Sec. 4.1); Then, we calibrate the episodic memory with the current query image using CT function (Sec. 4.2); Finally, we implement anomaly detection by measuring the distance of the calibrated memory and the current query image (Sec. 4.3). Moreover, we conduct analysis on the properties and complexity of our proposed model (Sec. 4.4).

## 4.1 Foreground-aware Semantic Memory

In practice, we may encounter the problem of anomaly detection for sparse foreground products. For the sake of algorithm effectiveness and computation efficiency, we propose to use pre-trained WRN and SAM to extract foreground and background features of support samples for constructing the semantic memory, as shown in Fig. 2. Specifically, following the existing studies [7, 31], we firstly use WRN parameterized by $\theta_{WRN}$ to extract the multi-scale feature maps of support images as $f_{1:k} = g_{\theta_{WRN}}(x_{1:k})$. Concurrently, we employ SAM with parameters $\theta_{SAM}$ to calculate masks of the corresponding support images and downsample them to the scale of support feature maps by $m_{1:k} = \text{ds}(g_{\theta_{SAM}}(x_{1:k}))$, where $\text{ds}(\cdot)$ is downsampling operation, $f_{1:k} \in \mathbb{R}^{k \times h \times w \times c}$ and $m_{1:k} \in \mathbb{R}^{k \times h \times w}$. Foreground and background features of support images can then be separately derived by:

$$f_{1:N_f}^{\text{fore}} = \text{fla}(f_{1:k} \times m_{1:k}), \quad f_{1:N_b}^{\text{back}} = \text{fla}(f_{1:k} \times (1 - m_{1:k})) \quad (3)$$

where $\text{fla}(\cdot)$ is a flatten operation. $N_f$ and $N_b$ are the number of feature vectors for the foreground and background, which are usually

huge. Hence, we compress the size of foreground and background features according to different downsampling ratios $\alpha$ and $\beta$ to reduce redundancy. Finally, the semantic memory can be represented by concatenation as $\mathcal{M}^s = [f_{1:\alpha N_f}^{\text{fore}}, f_{1:\beta N_b}^{\text{back}}] \in \mathbb{R}^{n \times c}$.

## 4.2 Episodic Memory Calibration with CT

Aware of the fact that query images are always coming online, we design an episodic memory calibration that is complementary to its semantic peer to better capture statistics of the current and historical query images. The episodic memory calibration is composed of memory augmentation and memory compression, the pipeline of which is shown in Fig. 3.

*4.2.1 Memory augmentation.* Memory augmentation aims to gather statistics of historical query images $x_{1:t-1}^q$ in the old episodic memory $\mathcal{M}_{t-1}^e$ with the features of the current query image $x_t^q$ to form the augmented memory $\mathcal{M}_t^e$, for the beginning we have $\mathcal{M}_0^e = \emptyset$. As shown in Fig. 3 (b), memory bank is defined as the concatenation of semantic and episodic memories and denoted by $\mathcal{M}_{t-1} = [\mathcal{M}^s, \mathcal{M}_{t-1}^e]$. We use the same WRN $g_{\theta_{WRN}}$ as in extracting features of semantic memory do for obtaining features of the current query image as $f_{1:m}^q = \text{fla}(g_{\theta_{WRN}}(x_t^q))$, where we replace the subscript t with q for simplicity. In order to obtain the new episodic memory augmented with statistics of the current query image, we need to solve the optimal matching flows between $\mathcal{M}^s$ and $f_{1:m}^q$. According to the analysis of CT theory in the preliminaries, we regard $\mathcal{M}^s$ and $f_{1:m}^q$ as two sets sampled from two distributions and formulate the distance with CT function as:

$$\mathcal{L}_{\phi,\rho} = \min_{\phi} \sum_{i=1}^{n} \sum_{j=1}^{m} \left[ \frac{\rho}{m} \pi_n(\mathcal{M}_i^s | f_j^q, \phi) c(\mathcal{M}_i^s, f_j^q) \right. $$
$$\left. + \frac{1-\rho}{n} \pi_m(f_j^q | \mathcal{M}_i^s, \phi) c(f_j^q, \mathcal{M}_i^s) \right] \quad (4)$$

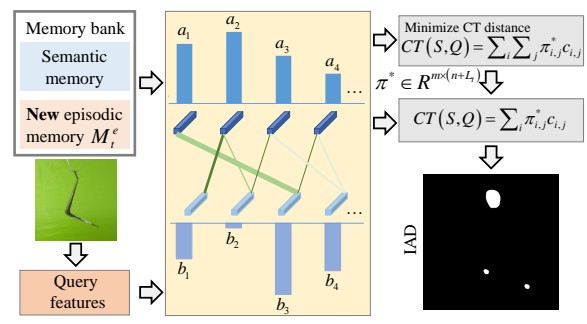

**Figure 4: Illustration on FS-IAD with our proposed FOCT.**

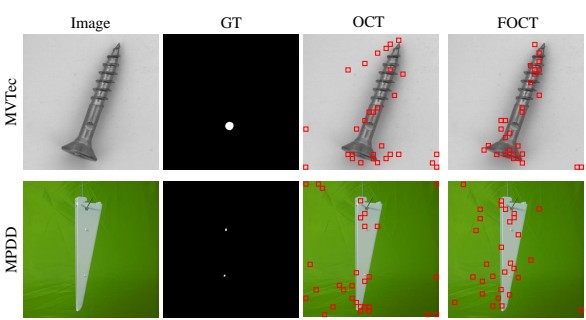

**Figure 5: Property of FOCT for sparse foreground products.**

For efficiently optimizing Eq. 4, we separately define the forward and backward transport matrices $\pi_n$ and $\pi_m$ as follows:

$$\pi_n(\mathcal{M}_i^s|f_j^q, \boldsymbol{\phi}) = \frac{exp(-d_{\boldsymbol{\phi}}(\mathcal{M}_i^s, f_j^q))}{\sum_{r=1}^n exp(-d_{\boldsymbol{\phi}}(\mathcal{M}_r^s, f_j^q))}$$

$$\pi_m(f_j^q|\mathcal{M}_i^s, \boldsymbol{\phi}) = \frac{exp(-d_{\boldsymbol{\phi}}(f_j^q, \mathcal{M}_i^s))}{\sum_{r=1}^m exp(-d_{\boldsymbol{\phi}}(f_r^q, \mathcal{M}_i^s))} \quad (5)$$

where $\boldsymbol{\pi}_n \in \mathbb{R}^{m \times n}$ and $\boldsymbol{\pi}_m \in \mathbb{R}^{n \times m}$. By minimizing Eq. 4 w.r.t. $\boldsymbol{\phi}$ via SGD algorithms such as Adam [18], Optimal flows between the semantic memory and current query features arrive.

Once the optimal forward transport matrix $\boldsymbol{\pi}_n^*$ is obtained, the memory augmentation with Top-K selection can be implemented:

$$\Delta \mathcal{M}_t^e = f_{idx}^q, \quad idx = TopK(argmax(\boldsymbol{\pi}_n^*, dim = 1)) \quad (6)$$

where $\Delta \mathcal{M}_t^e \in \mathbb{R}^{K \times c}$. It is worth noting that we conduct $argmax(\cdot)$ operation for each column since we want to select query features most relevant to normal features. Finally, the new episodic memory is refreshed by $\mathcal{M}_t^e = [\mathcal{M}_{t-1}^e, \Delta \mathcal{M}_t^e]$. The augmented memory bank can now be expressed by $\mathcal{M}_t = [\mathcal{M}^s, \mathcal{M}_t^e] \in \mathbb{R}^{(n+L_t) \times c}$, where $L_t$ is the length of episodic memory at the $t$-th query image.

*4.2.2 Memory compression.* In practice, the memory bank can not be infinite, suppose the capacity of the episodic memory is $L$. As the number of online query images increases, the episodic memory goes to full when $L_t \geq L$. Therefore, we design another mechanism called memory compression for efficient data management of episodic memory. The objective is to identify a subset $\overline{\mathcal{M}}_t^e \subset \mathcal{M}_t^e$ that the subset should enable the most accurate and expedited approximation of problem solutions over $\overline{\mathcal{M}}_t^e$, aligning closely with those computed over the entire $\mathcal{M}_t^e$. In our focus on anomaly detection, the upcoming subsection employs nearest neighbor computation. To guarantee comparable coverage of $\overline{\mathcal{M}}_t^e$ to the original episodic memory $\mathcal{M}_t^e$, we adopt a min-max facility location coreset selection [34], addressing the inherent NP-hard problem:

$$\overline{\mathcal{M}}_t^{e,*} = \underset{\overline{\mathcal{M}}_t^e}{argmin} \max_{u \in \mathcal{M}_t^e} \min_{v \in \overline{\mathcal{M}}_t^e} \|u - v\|_2^2 \quad (7)$$

where $\overline{\mathcal{M}}_t^{e,*} \in \mathbb{R}^{\lambda L \times c}$, $\lambda \in \{0, 1\}$ is compressive ratio. To solve the NP-hard problem, we employ an iterative greedy approximation [32]. To enhance the efficiency of memory compression, we use the Johnson-Lindenstrauss theorem proposed by [6] to reduce the

dimension of features within the episodic memory. After obtaining the compressed episodic memory, the new episodic memory can be depicted as $\mathcal{M}_t = [\mathcal{M}^s, \overline{\mathcal{M}}_t^{e,*}] \in \mathbb{R}^{(n+\lambda L) \times c}$.

## 4.3    Anomaly Detection

After parameters $\boldsymbol{\phi}^*$ of the CT function are optimized, the anomaly score map of the current query image can be expressed by:

$$s_j = \sum_{i=1}^n \pi_n(\mathcal{M}_i^s|f_j^q, \boldsymbol{\phi}^*) c(\mathcal{M}_i^s, f_j^q), \quad j = 1, ..., m \quad (8)$$

For image-level anomaly detection, we use the maximum distance score $s^*$ among all the pixels $s \in \mathbb{R}^m$ to represent $s^* = \max_{j=[1,m]} s_j$. For pixel-level localization, we first upscale the score map with bi-linear interpolation to match the original input resolution. Then, we smooth the score map with a Gaussian kernel width 4.

## 4.4    Model Analysis

*4.4.1 Property of FOCT on adaptive prior.* One of the important and appealing properties our proposed FOCT enjoys is that the prior distribution of query features can be adaptively acquired through the CT function optimization, which fits the practical scenario where the prior distribution of query features is usually unknown in advance. According to the forward transport matrix in Eq. 5, we have $\sum_{i=1}^n \pi_n(\mathcal{M}_i^s|f_j^q, \boldsymbol{\phi}^*) = 1$ while $\sum_{j=1}^m \pi_n(\mathcal{M}_i^s|f_j^q, \boldsymbol{\phi}^*) = C_i$, usually we have $C_i \neq 1$ and its real value is adaptive with query features and the procedure of CT function optimization.

*4.4.2 Property of FOCT for sparse foreground products.* Another practical and interesting property of our model is that it is computationally efficient, especially for sparse foreground products, which is usually encountered in real-world applications. As shown in Fig. 5, compared with its variant OCT without foreground-aware mechanism, FOCT shows more reasonable attention allocation in processing the foreground and background.

*4.4.3 Complexity analysis.* We implement $\boldsymbol{\phi}$ using a ReLU-activated MLP. Taking a two-layer MLP with dimensions $[d_1, d_2, d_3]$ for example. The forward and backward computational complexities per iteration take $O(NMd_2(d_1 + d_3))$, where $N$ and $M$ separately refer to the number of features of the two sets. To reduce the complexity, we can further control the size of MLP or the size of the two sets in achieving a trade-off between accuracy and real-time efficiency.

**Table 1: FS-IAD performance comparisons on MVTec and MPDD datasets. The results are averaged over all categories. Both image-level and pixel-level performances are reported in AUROC (%) ↑ and F1-max (%) ↑. The best results are in bold.**

| Dataset | Shot | PaDiM (ICPR'21) | | RegAD (ECCV'22) | | PatchCore (CVPR'22) | | FastRecon (ICCV'23) | | Ours | |
|---|---|---|---|---|---|---|---|---|---|---|---|
| | | AUROC | F1-max | AUROC | F1-max | AUROC | F1-max | AUROC | F1-max | AUROC | F1-max |
| MVTec | 1 | 76.6 / 89.3 | 88.2 / 40.2 | 82.9 / 92.5 | – / – | 76.7 / 81.6 | 88.5 / 38.9 | 84.9 / 93.3 | 91.6 / 49.2 | **87.1 / 94.4** | **91.6 / 51.7** |
| | 2 | 78.9 / 91.3 | 89.2 / 43.7 | 85.7 / 94.6 | – / – | 84.1 / 89.8 | 90.7 / 45.0 | 88.4 / 94.4 | 93.2 / 51.4 | **90.5 / 94.8** | **93.3 / 53.0** |
| | 4 | 80.4 / 92.6 | 90.2 / 46.1 | 88.2 / 95.8 | – / – | 88.5 / 91.2 | 92.6 / 48.1 | 91.2 / 96.0 | 94.1 / 52.9 | **93.2 / 96.2** | **94.9 / 54.9** |
| MPDD | 1 | 57.5 / 73.9 | – / – | 60.9 / 92.6 | – / – | 68.9 / 79.4 | 77.2 / 17.1 | 72.2 / **96.4** | 79.1 / 23.5 | **78.9** / 96.2 | **84.5 / 27.9** |
| | 2 | 58.0 / 75.4 | – / – | 63.4 / 93.2 | – / – | 75.5 / 84.4 | 81.7 / 23.4 | 76.1 / **96.7** | 82.8 / 28.8 | **82.4** / 96.5 | **86.7 / 28.9** |
| | 4 | 58.3 / 75.9 | – / – | 68.3 / 93.9 | – / – | 77.8 / 92.8 | 82.4 / 31.5 | 79.3 / **97.2** | 83.5 / 31.6 | **83.2** / 96.7 | **87.0 / 33.6** |

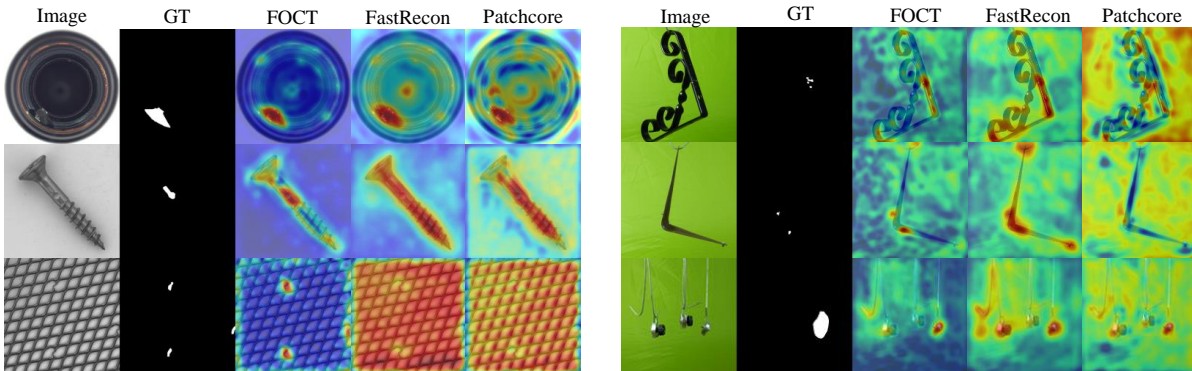

**Figure 6: Qualitative results of anomaly localization on MVTec (left) and MPDD (right) datasets under 2-shot scenario.**

## 5 EXPERIMENTS

### 5.1 Experimental Setups

**Dataset:** We conduct experiments on the MVTec [1] and MPDD [16] datasets. MVTec dataset comprises 5,354 images across 15 categories, with 3,629 defect-free and 1,725 defective images. Each category exhibits an average of five distinct defect types, with resolutions ranging from 700×700 to 1,024×1,024. MPDD dataset contains 6 classes of metal products. The images are captured from various distances and spatial angles amidst non-uniform backgrounds, which poses a significant challenge for the FS-IAD task. MPDD consists of 888 normal training images and 458 test images, which separately contain 176 normal images and 282 abnormal images. The image resolution is fixed as 1,024 × 1,024. Both datasets provide pixel-level ground truth annotations for defective regions.

**Competing Methods:** We compare our proposed FOCT against the most recently proposed SOTA full-data IAD and FS-IAD methods, including PaDiM[7], RegAD[14], PatchCore[28], and FastRecon[8]. For a fair comparison, we keep the query images fixed and conduct 10 random support image splittings under 1/2/4 shot scenarios. Then we use the official codes of PaDiM, RegAD, and PatchCore to implement experiments. We report the results of FastRecon with our own reproduction since the official code is not available yet.

**Evaluation Protocols:** To quantify the model performance of image-level anomaly detection, we employ the Area Under the Receiver Operator Curve (AUROC) and the F1-score at the optimal threshold (F1-max) as our evaluation metrics, being consistent with the previous works[15]. Furthermore, we utilize pixel-level AUROC

and F1-max to measure the defect localization performance. All the results of image-level and pixel-level AUROC and F1-max for competing methods and our model are averaged on the 10 splittings. **Implementation Details:** In our experiment, we employ a WRN-50 [41] pre-trained on ImageNet dataset [22] as the feature extractor. Image features are obtained from the intermediate layers following [28]. Both support and query images are reshaped to 336×336 in the MVTec and MPDD datasets. To initialize the semantic memory, foreground, and background subsampling ratios are $\alpha = 10\%$ and $\beta = 7.5\%$, respectively. Meanwhile, the episodic memory has a capacity of $L = 100$ feature vectors with a compression ratio $\lambda = 25\%$. We select the top 10 foreground and top 40 background feature vectors with the maximum transport probabilities. We update parameters of the CT function using Adam[18] with a learning rate of 0.0001 for 100 epochs, with the trade-off factor of CT $\rho = 0.5$. All the experiments are conducted on one NVIDIA GTX 3090 GPU. We will release the source code upon acceptance.

### 5.2 Comparisons with SOTA Methods

*Few-shot Anomaly Detection.* We compare the few-shot anomaly detection performance of our proposed FOCT with recently proposed competitive baselines and report image-level results with black in Table 1. Our model significantly outperforms the previous methods on both MVTec and MPDD datasets under image-level AUROC and image-level F1-max metrics. Compared with the results of PaDiM and PatchCore, taking 1-shot for example, AUROC and F1-max of our model separately exceed the two baselines more

**Table 2: Anomaly detection comparisons per product in image-level and pixel-level AUROC (%) ↑ on MVTec and MPDD datasets under 2-shot setting. The best results are in bold.**

|  | Dataset | PatchCore (CVPR'22) | FastRecon (ICCV'23) | Ours |
|---|---|---|---|---|
| MVTec | Bottle | 99.0 / 94.3 | **100.0** / **98.5** | 99.4 / 98.3 |
|  | Cable | 81.7 / 87.0 | 83.2 / 94.5 | **85.0** / **94.6** |
|  | Capsule | 74.6 / 88.5 | 77.0 / **98.2** | **81.5** / 97.6 |
|  | Hazelnut | 98.8 / 96.4 | **99.3** / **97.8** | 98.8 / 92.5 |
|  | MetalNut | 66.7 / 82.4 | 92.0 / **96.8** | **95.1** / 93.8 |
|  | Pill | 79.8 / 86.9 | **93.0** / **98.1** | 92.1 / 94.4 |
|  | Screw | 43.7 / 89.9 | 44.4 / 91.9 | **54.8** / **95.0** |
|  | Toothbrush | 78.3 / 93.5 | 78.1 / 94.2 | **83.1** / **96.1** |
|  | Transistor | 72.2 / 72.5 | 84.2 / **89.8** | **88.2** / 79.4 |
|  | Zipper | 95.9 / 97.1 | 97.7 / **98.7** | 97.2 / **98.7** |
|  | Carpet | 97.0 / 98.8 | **99.5** / **99.2** | 98.1 / 99.0 |
|  | Grid | 75.9 / 70.5 | 79.9 / 67.8 | **87.2** / **91.9** |
|  | Leather | **100.0** / **99.4** | **100.0** / 99.3 | **100.0** / **99.4** |
|  | Tile | 99.1 / 95.9 | **99.3** / **97.0** | 99.2 / 95.5 |
|  | Wood | **98.7** / 94.8 | 98.4 / 94.6 | 98.5 / **94.9** |
|  | Average | 84.1 / 89.9 | 88.4 / 94.4 | **90.5** / **94.7** |
| MPDD | Bracketblack | 64.7 / 94.1 | 63.3 / **96.8** | **71.0** / 96.4 |
|  | Bracketbrown | 51.8 / 65.8 | 52.3 / **93.4** | **56.1** / 92.8 |
|  | Bracketwhite | 74.0 / 96.2 | 64.1 / **97.0** | **81.9** / 95.9 |
|  | Connector | 85.9 / 89.1 | 91.4 / 96.4 | **99.0** / **97.5** |
|  | Metalplate | 99.0 / 91.3 | **100.0** / **98.8** | 97.8 / 98.6 |
|  | Tubes | 77.8 / 70.1 | 85.6 / **97.8** | **88.3** / 97.6 |
|  | Average | 75.5 / 84.4 | 76.1 / **96.7** | **82.4** / 96.5 |

**Table 3: Ablation studies with image-level and pixel-level metrics on MPDD under 2-shot case. The best results are in bold.**

| CT | Online | Foreground | AUROC | F1-max |
|---|---|---|---|---|
| - | - | - | 76.1 / **96.7** | 82.8 / 28.8 |
| ✓ | - | - | 77.8 / 96.6 | 84.9 / 29.7 |
| ✓ | ✓ | - | 77.1 / 96.4 | 85.3 / **31.4** |
| ✓ | ✓ | ✓ | **82.4** / 96.5 | **86.7** / 28.9 |

than 4% and 3%, which reveals the fact that transductive inference in few-shot scenario benefits image-level anomaly detection a lot since more statistics on query images are considered. When it comes to results between FastRecon and our FOCT, also considering the 1-shot case, we observe that our model outperforms FastRecon more than 2% in AUROC. We attribute this to the fact that our model simultaneously employs powerful non-linear measurement functions and leverages statistics of query images more thoroughly. It is worth noting that products' sparsity in MPDD is more obvious than MVTec, and our model achieves more gain on MPDD against MVTec. It verifies the effectiveness of our proposed foreground-aware semantic memory to some extent. Specifically, our FOCT exceeds the second-best model by more than 6% and 5% in image-level AUROC and image-level F1-max under 1-shot setting. Qualitative results are reported in Fig. 6.

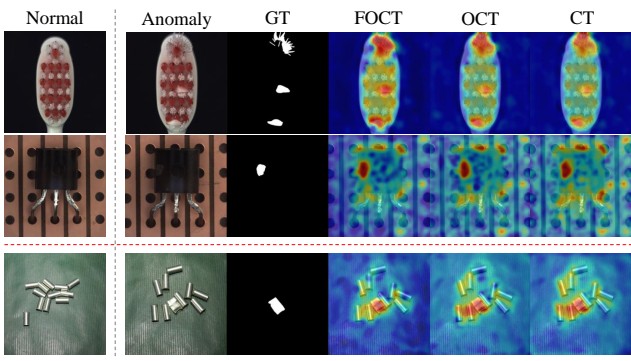

**Figure 7: Qualitative ablation studies of 2-shot anomaly localization on MVTec (top) and MPDD (bottom) datasets.**

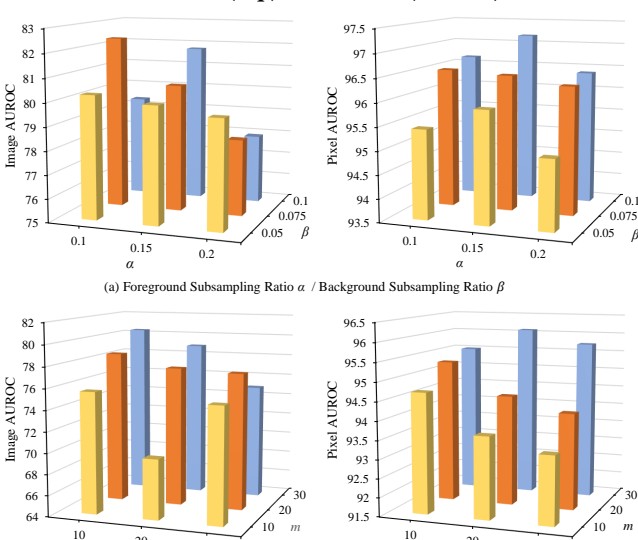

(a) Foreground Subsampling Ratio $\alpha$ / Background Subsampling Ratio $\beta$

(b) Foreground Selection number $n$ / Background Selection number $m$

**Figure 8: Image-level and pixel-level AUROCs (%) ↑ of 2-shot on MPDD dataset versus foreground and background (a) subsampling ratio; (b) selection number.**

*Few-shot Anomaly Localization.* We report the few-shot anomaly localization results of our model compared with other competitive baselines using pixel-level AUROC and pixel-level F1-max with gray in Table 1. Our model outperforms the competitors on the MVTec dataset across the two metrics under various few-shot scenarios including 1/2/4 shot. Our model achieves competitive pixel-level AUROC and exceeds more than 2% on average using pixel-level F1-max against other competitors. An interesting observation is that our model does not achieve SOTA performance in pixel-level AUROC, which we attempt to answer hereafter. We find that the ground truth masks of MPDD datasets are sometimes larger than the actual anomaly areas, examples can be found in Fig. 6. However, anomaly score maps derived from our model have low uncertainty. Therefore, the predicted mask after binarizing the score map may have some relatively concentrated anomalous areas. When calculating the metric of pixel-level AUROC, although our model precisely localizes the anomalous areas, the IOU between

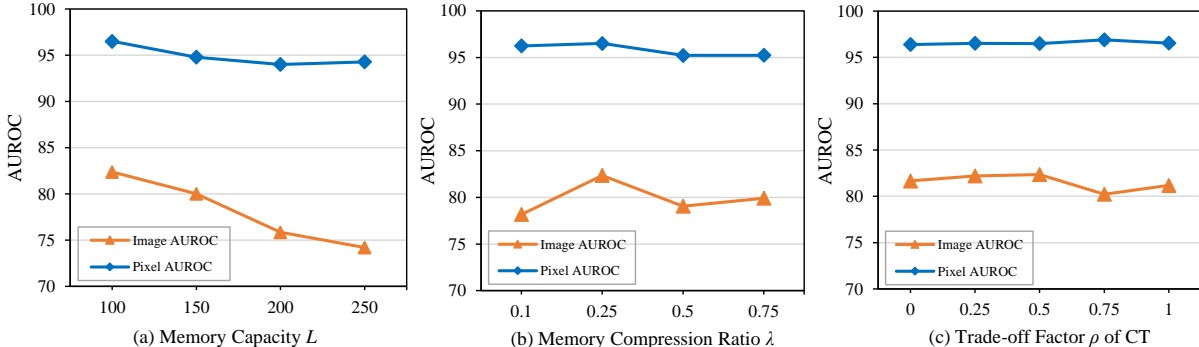

**Figure 9: Image-level and pixel-level AUROCs (%) ↑ on MPDD dataset under 2-shot setting versus (a) the memory capacity** L**, (b) the memory compression ratio** $\lambda$**, and (c) the trade-off factor** $\rho$ **of CT function.**

the human-annotated ground truth mask and the concentrated predicted mask remains low, which negatively impacts the recall value of our model and leads to lower pixel-level AUROC. Besides, we provide performance comparisons per product and report the results in Table 2 for more detailed comprehension.

## 5.3 Ablative Analysis

In our ablation study, we develop three variants of our proposed FOCT called vanilla, CT, and OCT to evaluate the effectiveness of different components of our model. Compared with FOCT, OCT removes the foreground-aware mechanism in semantic memory and only uses downsampling to construct the semantic memory. Compared with OCT, CT only leverages statistics of the current query image and ignores statistics of historical query images. Compared with CT, vanilla uses statistics of the current query image via the linear measurement function introduced in FastRecon [8]. The results are reported in Table 3 and Fig. 7, respectively.

*Affect of non-linear measurement function.* Comparing the results between vanilla and CT in Table 3, we observe that CT significantly outperforms vanilla in almost all metrics. It approves the superiority and effectiveness of non-linear measurement powered by the CT function against its linear measurement peer.

*Affect of statistics of historical query images.* Comparing the results between CT and OCT, OCT achieves consistent improvement in general. It reveals that statistics of historical query images matter in anomaly detection when the normal training images are extremely limited. We also find that the metric of AUROC drops a little in the OCT variant, and we attribute this to that OCT augments too many redundant backgrounds into the memory bank, thus restricting anomaly detection performance in low-data regime.

*Affect of foreground-aware semantic memory.* According to the results between OCT and FOCT, there exhibits some significant improvements in image-level metrics of AUROC and F1-max, demonstrating the necessity and effectiveness of foreground-aware semantic memory, qualitative visualizations can be found in Fig. 5.

## 5.4 Hyper-parameters Analysis

In this part, we mainly focus on analyzing the robustness of our model against various hyper-parameters, including foreground

and background subsampling ratios $\alpha$ and $\beta$, foreground and background selection numbers n and m, memory capacity L, memory compression ratio $\lambda$, and trade-off factor $\rho$ of CT function. Results are separately reported in Fig. 8 and Fig. 9.

*Affect of memory initialization.* We regard $\alpha$, $\beta$, n, and m mentioned above as hyper-parameters of memory initialization. As we can see an initialized memory bank with a proper foreground and background ratio is crucial for generalized anomaly detection in the low-data regime. We attribute this to the fact that the representativeness of the initial memory is related to such a ratio.

*Affect of memory capacity.* We observe that the oversized memory capacity with large L is unfriendly with boosting performance. We believe that unexpected interference would be introduced with the increase in memory capacity, which should have negative impacts on anomaly detection.

*Affect of memory compression ratio.* As we can see, generally speaking, the low or high compression ratio $\lambda$ hurts AUROCs. On one hand, the low $\lambda$ may sacrifice the diversity of the memory bank. On the other hand, the high $\lambda$ may not fully utilize the statistics of the current query images.

*Affect of trade-off factor in CT function.* Our model achieves robust AUROCs versus various $\rho$ values, demonstrating the flexibility of our model when learning the CT function. Besides, the best AUROCs arrive at $\rho = 0.5$, which implies that both the forward and backward CTs are important for improving performance.

## 6 CONCLUSION

In this work, we propose a novel memory-based TFS-IAD framework, called FOCT, which has rarely been studied before. It consists of three components, including foreground-aware semantic construction, episodic memory calibration, and anomaly detection in the few-shot scenario. The episodic memory calibration and anomaly detection are both powered by the non-linear measurement CT function, compared with its linear counterparts without considering foreground sparsity and statistics of online query images, FOCT shows impressive generalization capability across various benchmark datasets under fruitful few-shot scenarios of 1/2/4 shot.

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
