# OpenReview forum: "FOCT: Few-shot Industrial Anomaly Detection with Foreground-aware Online Conditional Transport"
_acmmm.org/ACMMM/2024/Conference — MM2024 Poster_

### Official Review · Reviewer_6cy5 · 2024-04-30

**Rating:** 4
**Confidence:** 3

**Summary:**

This paper propose a memory-based transductive few-shot industrial anomaly detection framework. This framework adopts a non-linear conditional transport (CT) function between the memory bank and query images for precise feature matching. A pipeline called FOCT is designed to enhance the generalization capability of the aforementioned CT function.

**Strengths:**

1. This paper propose a memory-based TFS-IAD framework that primarily adopts a non linear and theoretical guaranteed CT function between the memory bank and query images for precise feature matching;
2. A pipeline called FOCT is designed to enhance the generalization capability of the aforementioned CT function;
3. Compared with other competing methods, the performance of the proposed methods has improvement.

**Limitations:**

1. The standard deviation of the results of multiple experiments is not reported;
2. The number of competing methods is suggested to increase, and currently there are only four competing methods;
3. The review about AD should be added in the related work section;
4. Insufficient description of foreground-aware semantic memory methods. In section 4.1, please explain why it is necessary to downsample the foreground and background with two different proportions of α and β. The previous content seems to mention only that the foreground and background have huge sizes.

**Suitability:**

2

---

### Official Review · Reviewer_YdcY · 2024-05-25

**Rating:** 3
**Confidence:** 4

**Summary:**

This paper proposes a novel memory-based TFS-IAD framework that primarily adopts a non-linear and theoretical guaranteed CT function between the memory bank and query images for
precise feature matching. This framework can be plugged and played with any pre-trained backbones, such as WRN and SAM. The effectiveness is demonstrated on MVTec-AD and MPDD.

**Strengths:**

This paper focuses on online/transductive anomaly detection, which is a promising and practical perspective.

**Limitations:**

1. The writing and organization are somewhat confusing and can be largely polished.
2. Some contents are missing/unclear, e.g., what the \delta in line 325-326 means? And for lines 628-629: “For a fair comparison, we keep the query images fixed and conduct 10 random support image splittings under 1/2/4 shot scenarios.” Can you provide a clearer explanation?
3. I also question the process of “episodic memory calibration”. In my opinion, the anomaly percentage of the query set is unknown, so how are the two hyper-parameters 10 and 40 in the “top 10 foreground and top 40 background feature” determined? And their ablation studies are missing.
4. To date, the results are not convincing. A series of important baselines are missing, e.g., GraphCore, WinCLIP, etc. Additionally, for the screw category of the MVTec-AD dataset, the AUC result does not show obvious improvement in the TFS setting. Some analysis is necessary.
5. To my understanding, the F1-max only depends on the anomaly score. So why this evaluation is missing in PaDim on MPDD and RegAD on MVTec-AD and MPDD?
6. For IAD, the efficiency of inference time also matters but the authors do not mention that.

**Suitability:**

1

---

### Official Review · Reviewer_Xpsa · 2024-05-29

**Rating:** 4
**Confidence:** 2

**Summary:**

To address the matching issue between queries and supports in Few-Shot Industrial Anomaly Detection, the authors proposed using Conditional Transport (CT) as a metric to compute the structural distance between two sets of representations to determine feature relevance. Additionally, they dynamically update the memory bank by utilizing SAM for foreground segmentation. Extensive experiments on various FS-IAD benchmarks demonstrate the performance of the proposed method.

**Strengths:**

1. The feature visualizations of the experiments are reasonable.
2. Conditional Transport is used for the first time in anomaly detection.

**Limitations:**

1. The author mentioned that FOCT is flexible and a Plug-and-Play Module, but there is no additional experimental evidence to prove its effectiveness in different scenarios.
2. The online method is used for updating the memory bank. However, the results in Table 3 show little improvement.
3. The random selection of support samples may lead to significant variations in the results under the low-data regime. It is recommended to select multiple sets of support samples and calculate the mean and standard deviation.
4. What does the red box in Figure 5 mean?
5. How is old episodic memory acquired, and is it the same feature format as semantic memory?
6. For certain objects like Carpet, Leather, Tile, and Wood, how to utilize SAM for foreground and background separation. Lacks some specific examples for demonstration

**Suitability:**

2

---

### Official Review · Reviewer_N7vA · 2024-06-05

**Rating:** 5
**Confidence:** 2

**Summary:**

This research paper presents a framework for addressing Few-Shot Industrial Anomaly Detection (FS-IAD), which is critical in scenarios where data efficiency and the ability to quickly adapt across different products are prioritized. The framework utilizes pre-trained models for initial image segmentation and employs Optimal Matching Flows, Conditional Transport (CT), and Online Conditional Transport (OCT) to dynamically calibrate and match memory banks with query images in real-time. This approach is particularly tailored for scenarios where query images vary significantly and are processed either individually or in batches. The model also introduces Foreground-aware OCT (FOCT) to enhance focus on diverse targets over redundant backgrounds during memory calibration, which is beneficial for sparse foreground settings.

**Strengths:**

1.Enhanced Data Efficiency: The model is designed to work effectively with very few samples per category, making it ideal for applications where data is scarce or expensive to obtain.
2.Adaptability: The framework supports rapid adaptation to new product lines or changes in production scenarios without requiring extensive retraining, thanks to its Few-Shot learning approach.
3.Dynamic Calibration: By using OCT, the model continually updates its memory bank with new query images, allowing for real-time anomaly detection and improved responsiveness to new abnormalities.
4.Focus on Relevant Features: FOCT improves the model’s efficiency by emphasizing significant foreground elements, which is crucial for detecting anomalies in environments with sparse foreground data.
5.State-of-the-Art Performance: Demonstrated effectiveness across diverse datasets, achieving state-of-the-art results, which indicates its robustness and reliability in industrial settings.

**Limitations:**

1.The framework's performance heavily relies on the quality of the pre-trained segmentation and backbone models, which might limit its effectiveness if not properly tuned or if unsuitable models are used.
2. The use of advanced techniques like CT and OCT may introduce complexity in implementation and require significant computational resources, especially in real-time scenarios.
3.The introduction of pre-trained models and SAM etc. has increased the complexity and computational requirements, and the authors should probably give more relevant analyses.
4.For addressing few-shot problems, it is beneficial for researchers to validate their algorithms using more robust and challenging datasets. An exemplary dataset for such evaluations is the Real-iad: A real-world multi-view dataset designed specifically for benchmarking versatile industrial anomaly detection. This dataset provides a practical and comprehensive testing ground to assess the effectiveness and robustness of new algorithms under real-world conditions.

**Suitability:**

3

---

### Meta-Review · Area_Chair_4Ce7 · 2024-07-01

**Recommendation:** Accept (Poster)
**Confidence:** 4

**Metareview:**

This submission presents a novel approach to few-shot industrial anomaly detection (FS-IAD) by introducing a framework that leverages Conditional Transport (CT) and dynamic memory bank updates using SAM for foreground segmentation. The proposed method is evaluated on various FS-IAD benchmarks, demonstrating its potential effectiveness.

Pros:
- The use of Conditional Transport as a metric for determining feature relevance is novel in anomaly detection.

- The approach of dynamically updating the memory bank for foreground segmentation is interesting.

Cons:

- The lack of competing methods was mentioned by reviewers. During the rebuttal, authors presented additional comparisons with GraphCore and WinCLIP, which eased the concerns.

- The random selection of support samples can lead to significant variations in results under low-data conditions. During rebuttal, authors included additional results demonstrating the stability of resulting by taking the average and standard deviation over multiple runs.

Overall, this paper presents a promising approach to few-shot industrial anomaly detection with some innovative ideas. However, there are several areas that require further clarification and additional experiments to strengthen the claims. Given the current ratings (2 weak accepts, 1 borderline accept, and 1 borderline reject), I would recommend to accept this submission.